# SurstSplat: Dynamic Surgical Gaussian Reconstruction via Spatiotemporal Graph Matching

## Abstract

Reconstructing dynamic 3D models from surgical videos is crucial for advanced medical applications, but faces challenges from limited textures, inconsistent lighting, and complex tissue deformations. We present *SurstSplat*, a framework for dynamic Gaussian reconstruction with spatiotemporal semantic graph matching. By integrating multimodal features from pre-trained 2D foundation models into 3D Gaussian representations, our approach aims to better capture tissue deformations and tool interactions. The proposed graph matching regularizes semantics across space and time and supports downstream tasks such as semantic segmentation, language-guided editing, and medical visual question answering. In experiments on endoscopic datasets, we observe improvements in rendering quality and semantic performance under challenging conditions, while maintaining competitive efficiency.

## 1 Introduction

High-quality reconstruction of surgical videos is essential in modern medicine, providing anatomical precision for planning, supporting real-time decision-making, and creating immersive training environments. While traditional methods such as structured light and SLAM Fuentes-Pacheco et al. (2015); Kazerouni et al. (2022) are common, they struggle with soft tissue geometries and dynamic surgical environments, resulting in inaccuracies and incomplete reconstructions. Though volumetric approaches show potential, they remain challenged by deformable tissue representation under variable lighting conditions, necessitating the development of advanced techniques that combine precision with real-time capabilities.

Recent advances in Neural Radiance Fields (NeRF) have enabled 3D surgical scene reconstruction from multi-view images, but their computational demands limit real-time use. EndoNeRF captures tissue deformations with canonical fields and time-dependent displacements, yet remains too slow for surgical environments. 3D Gaussian Splatting (GS) Kerbl et al. (2023b) provides a faster, differentiable point cloud approach for accelerated rendering.

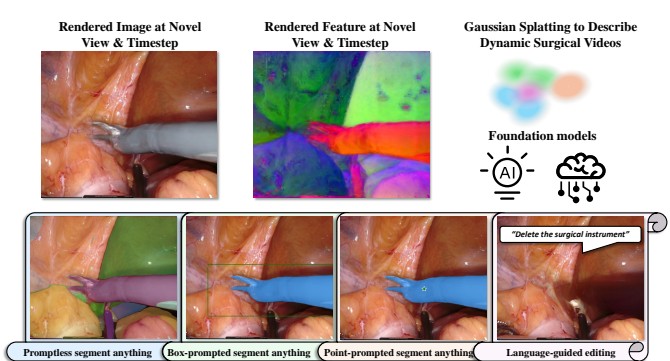

Figure 1: **Overview of *SurstSplat*:** We embed multimodal features from pre-trained 2D models into 3D Gaussian Splatting, enhancing surgical scene reconstruction and enabling advanced visualization for robot-assisted surgery.

Building on this, methods like EndoGaussian Liu et al. (2024b) and EndoGS Zhu et al. (2024) use Gaussian parameterization and SLAM-based techniques to improve deformable tissue modeling

and camera pose estimation. These advances enable efficient 3D reconstruction from 2D videos, supporting surgical planning.

Despite significant advancements in surgical scene reconstruction techniques Liu et al. (2024b); Yang et al. (2024), two key challenges remain for practical medical use. First, although medical reconstructions are often technically successful, they are difficult for non-professionals to interpret due to the need for specialized expertise, which hinders broader adoption in clinical settings where interdisciplinary communication is essential Rodríguez et al. (2022); Liu et al. (2024a). Current 4D Gaussian models also struggle to support critical downstream tasks such as real-time scene editing, semantic understanding, and automatic surgical report generation, limiting their utility in comprehensive surgical workflows Mahmoud et al. (2017).

Second, dynamic surgical environments present unique challenges including limited surface textures, inconsistent lighting conditions Batlle et al. (2023); Yang et al. (2023), and continuous tissue deformations. Traditional reconstruction methods struggle to achieve sufficient rendering quality in areas with smooth tissue surfaces or complex reflections. The dynamic nature of surgical scenes—characterized by tissue deformations from physiological processes and interactions with surgical instruments—significantly increases reconstruction complexity Liu et al. (2024b). These spatiotemporal dynamics require specialized approaches that capture geometric changes while maintaining semantic consistency across frames.

To address these challenges, we propose SurstSplat, a framework that integrates spatiotemporal semantic graph matching into dynamic Gaussian reconstruction. By embedding multimodal feature fields from pre-trained 2D foundation models into 3D Gaussian representations, our approach encourages temporal semantic consistency and supports handling of deformable tissues and tool interactions. We introduce a spatiotemporal graph distillation mechanism that establishes semantic correspondences across frames. On endoscopic datasets, we observe improved rendering quality and semantic performance under challenging conditions. Conceptually, our goal is to provide surgeons and clinicians with a temporally stable, semantically annotated 4D representation that improves spatial understanding of anatomy and tools from arbitrary viewpoints, which— to the best of our knowledge—has not been demonstrated before with dynamic Gaussian splatting. Our contributions are summarized as:

- We introduce *SurstSplat*, integrating feature fields from pre-trained foundation models into dynamic GS to jointly model appearance and semantics for surgical reconstruction.
- We propose a spatiotemporal semantic graph matching mechanism that regularizes semantics across space and time and complements the deformation field; ablations indicate correlation with improved temporal coherence.
- We show that *SurstSplat* supports promptable segmentation, language-guided editing, and medical visual question answering, with competitive FPS on our setup.
- Compared with prior feature-augmented Gaussian splatting works such as Feature 3DGS, LangSplat, and 4D LangSplat, which mainly target static or generic scenes, *SurstSplat* is tailored to the clinical setting: it couples dynamic 4D Gaussians with an explicit spatiotemporal semantic graph, enabling zero-shot editing and VQA on deformable surgical videos.

## 2 METHOD

We introduce *SurstSplat*, a dynamic Gaussian Splatting framework for clinical scene reconstruction that fuses multimodal feature-based rendering with feature field distillation. Our approach tackles two main challenges: poor rendering quality due to limited textures and photometric inconsistencies, and the demand for advanced semantic understanding. By embedding multimodal features from pre-trained 2D foundation models into the Gaussian radiance field, *SurstSplat* enables unified radiance and semantic representation, supporting applications like 3D tissue visualization and medical visual question answering. Figure 2 shows the architecture of *SurstSplat*.

### 2.1 TEMPORAL-AWARE DYNAMIC GAUSSIAN SPLATTING

Gaussian Splatting Kerbl et al. (2023b) utilizes a set of dense Gaussians to represent 3D data and achieve real-time rendering of scenes. Each Gaussian $\Theta$ is defined by its center $\boldsymbol{\mu} \in \mathbb{R}^3$, covariance

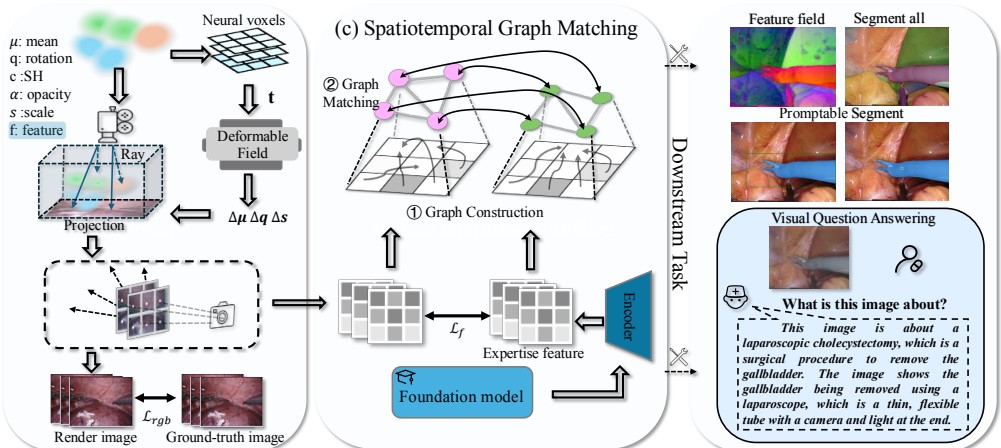

(a) Dynamic Gaussian Splatting  (b) Semantics-embedded Dynamic Rendering  (d) Feature-prompted 3D Descriptor

Figure 2: **Overview of Proposed *SurstSplat*.** (a) Temporal-Aware Dynamic Gaussian Splatting projects deformable neural voxels into 3D scenes. (b) Semantics-embedded Dynamic Feature Rendering integrates foundation model priors for improved feature extraction. (c) Spatiotemporal Graph Matching aligns features across frames. (d) The resulting promptable scene representation supports various downstream applications.

matrix $\boldsymbol{\Sigma} \in \mathbb{R}^{3\times 3}$ (decomposed into a scaling factor $\boldsymbol{s} \in \mathbb{R}^3$ and a rotation quaternion $\boldsymbol{q} \in \mathbb{R}^4$), opacity $\sigma \in \mathbb{R}$, and SH coefficients $\boldsymbol{\alpha} \in \mathbb{R}^C$ for colors and view-dependent appearance. The Gaussian function is expressed as:

$$G(\boldsymbol{x}) = e^{-\frac{1}{2}(\boldsymbol{x}-\boldsymbol{\mu})^T \boldsymbol{\Sigma}^{-1}(\boldsymbol{x}-\boldsymbol{\mu})}, \tag{1}$$

For static scenes, the attributes of the i-th Gaussian are defined as $\Theta_i = \{x_i, q_i, s_i, \sigma_i, c_i\}$. The rendering process follows the differentiable 3D Gaussian splatting framework, where the color $C$ of a pixel is computed using volumetric rendering:

$$C = \sum_{i \in \mathcal{N}} c_i \, \sigma_i \prod_{j=1}^{i-1} (1 - \sigma_j), \tag{2}$$

where $\mathcal{N} = \texttt{overlap}(M, \mu, q, s)$ is the set of 3D Gaussians overlapping the given pixel, determined by the view matrix $M = [R, T]$. To handle dynamic scenes, the framework is extended by integrating temporal information Wu et al. (2023). A spatial-temporal feature field $\mathcal{E}$ and an MLP decoder $\mathcal{D}$ model the deformation of each Gaussian over time. Given the Gaussian center $\boldsymbol{\mu} = (x, y, z)$ and a query time $t$, the deformation in position, rotation, and scaling is computed, and the Gaussian parameters are then updated:

$$\Delta\boldsymbol{\mu}, \Delta\boldsymbol{\omega}, \Delta\boldsymbol{s} = \mathcal{D}(\mathcal{E}(\boldsymbol{\mu}, t)), \tag{3}$$

$$\boldsymbol{\mu}', \boldsymbol{q}', \boldsymbol{s}' = \boldsymbol{\mu} + \Delta\boldsymbol{\mu}, \; \text{Normalize}\big(\text{Exp}(\Delta\boldsymbol{\omega}) \otimes \boldsymbol{q}\big), \; \boldsymbol{s} + \Delta\boldsymbol{s}. \tag{4}$$

Here $\otimes$ denotes quaternion multiplication. $\text{Exp}(\cdot)$ maps a 3D axis–angle vector to a unit quaternion via the exponential map, and $\text{Normalize}(\cdot)$ ensures $\|\boldsymbol{q}'\| = 1$.

**Deformation Field.** We parameterize the deformation field $\mathcal{D} : (\boldsymbol{x}, t) \mapsto (\Delta\boldsymbol{\mu}, \Delta\boldsymbol{\omega}, \Delta\boldsymbol{s})$ with an MLP that takes spatial coordinates, a time encoding (e.g., sinusoidal $\gamma(t)$), and optionally a local semantic feature $\boldsymbol{f}_{(l,t)}$ to predict translation, rotation (axis–angle), and scale residuals. Rotations use the exponential map to ensure unit quaternions, and the field is integrated with graph matching to enforce temporally consistent correspondences across frames. Importantly, the deformation field remains the sole component that updates the Gaussians' geometric state $(\boldsymbol{\mu}', \boldsymbol{q}', \boldsymbol{s}')$; the spatiotemporal graph only regularizes the associated semantic features and never directly modifies geometry, so it complements rather than replaces the underlying dynamic GS model.

**Neural Voxels.** We use a coarse 3D grid of learnable features ("neural voxels") trilinearly sampled at Gaussian centers, which condition the deformation MLP and semantic projection to provide

spatial context and enhance robustness in texture-poor surgical regions. Intuitively, these voxels act as a low-resolution latent volume that encodes local anatomical context (e.g., organ vs. background), which improves stability of the deformation prediction in regions where RGB cues alone are unreliable.

For dynamic scenes, the rendering process applies volumetric rendering to these dynamically updated Gaussians, allowing for real-time adaptation of the scene representation to account for movements and changes in the environment:

$$C' = \sum_{i \in \mathcal{N}'} c_i \, \sigma_i \prod_{j=1}^{i-1} (1 - \sigma_j), \tag{5}$$

where $\mathcal{N}' = \texttt{overlap}(M, \mu', q', s')$ is set of updated dynamic Gaussians overlapping given pixels.

## 2.2 SEMANTICS-EMBEDDED DYNAMIC FEATURE RENDERING

Building upon the basic Gaussian Splatting, we incorporate semantic features into the dynamic 3D representation. Each Gaussian $\Theta_i$ now includes a semantic feature vector $\boldsymbol{f}_i \in \mathbb{R}^N$, where $N$ is the latent dimension. These features, derived from 2D foundation models like SAM, CLIP-LSeg, and LLAVA-Med, capture semantic information crucial for tasks such as 3D segmentation, language-guided editing, and medical visual question answering. Given a view matrix $M = [R, T]$, we extend the rendering process to handle high-dimensional feature maps. The semantic feature value $F$ of a pixel is computed using volumetric rendering:

$$F = \sum_{i \in \mathcal{N}} f_i \, \sigma_i \prod_{j=1}^{i-1} (1 - \sigma_j), \tag{6}$$

where $\mathcal{N} = \texttt{overlap}(M, \mu, q, s)$ is the set of Gaussians overlapping the given pixel. This integration enables simultaneous rendering of RGB images and semantic maps in 3D space, with semantic features optimized alongside geometric and appearance attributes.

For dynamic scenes, each Gaussian is augmented with a semantic feature vector $\boldsymbol{f}_i \in \mathbb{R}^N$ and a temporal component $t$, capturing spatial, semantic, and temporal cues. Multi-modal features from foundation models (e.g., SAM Kirillov et al. (2023), LLAVA-Med Li et al. (2024a)) are distilled into these dynamic Gaussians, enabling real-time rendering of RGB images and semantic maps with strong temporal consistency.

Given the view matrix $M = [R, T]$ and timestamp $t$, we extend differentiable Gaussian splatting to support high-dimensional features and temporal dynamics. The original 3D Gaussians $\mathcal{G}$ are converted into time-dependent Gaussians $\mathcal{G}'(t)$, and volumetric rendering computes both semantic feature $F'$ and color $C'$ for each pixel, naturally reflecting temporal changes.

$$C' = \sum_{i \in \mathcal{N}'} c_i \, \sigma_i \prod_{j=1}^{i-1} (1 - \sigma_j), \quad F' = \sum_{i \in \mathcal{N}'} f_i \, \sigma_i \prod_{j=1}^{i-1} (1 - \sigma_j), \tag{7}$$

where $\mathcal{N}'$ is the set of 4D Gaussians overlapping the pixel, i.e., $\mathcal{N}' = \texttt{overlap}(M, \mu', q', s')$, with $c_i$, $f_i$, and $\sigma_i$ as time-dependent color, feature, and opacity. Time conditioning enables per-frame updates of semantic and spatial representations, supporting real-time tasks. Supervision with sequential feature maps $F(t)$ from pre-trained 2D models teaches spatiotemporal semantics for consistent, high-quality dynamic scene rendering.

## 2.3 SPATIOTEMPORAL GRAPH MATCHING FOR SEMANTICS TRANSFER

To address the unique challenges of clinical videos, including rapid camera motion and complex tissue deformations, we introduce a spatiotemporal graph distillation mechanism. This approach is based on the insight that while 2D dynamics in the video might be complex, the underlying 3D motion is often low-dimensional and composed of simpler units of rigid motion.

**Spatiotemporal Graph Construction.** We formalize the spatiotemporal graph as $\mathcal{G} = (\mathcal{V}, \mathcal{E})$. Each node $v_{(l,t)} \in \mathcal{V}$ corresponds to a Gaussian $g_{(l,t)}$ at spatial index $l$ and time $t$, with pose and appearance $(\boldsymbol{x}_{(l,t)}, \boldsymbol{q}_{(l,t)}, \boldsymbol{s}_{(l,t)}, \sigma_{(l,t)})$ and semantic feature $\boldsymbol{f}_{(l,t)}$. Edges comprise within-frame spatial

links and cross-frame temporal links: $\mathcal{E} = \mathcal{E}_{\text{space}} \cup \mathcal{E}_{\text{time}}$. Spatial edges connect $k$-NN neighbors in 3D (or 2D projection) within the same frame; temporal edges connect nodes across a window $t' \in [t - \Delta t, t + \Delta t]$ using top-$k$ matches by semantic similarity. The edge weight between $u = (l, t)$ and $v = (l', t')$ is

$$w_{uv} = \underbrace{\exp\left(-\frac{\|\boldsymbol{x}_u - \boldsymbol{x}_v\|^2}{\tau_x^2}\right)}_{\text{spatial}} \cdot \underbrace{\frac{\langle \hat{\boldsymbol{f}}_u, \hat{\boldsymbol{f}}_v \rangle}{\|\hat{\boldsymbol{f}}_u\| \, \|\hat{\boldsymbol{f}}_v\|}}_{\text{semantic}} \cdot \underbrace{\exp\left(-\frac{|t - t'|}{\tau_t}\right)}_{\text{temporal}}, \tag{8}$$

with temperatures $\tau_x, \tau_t > 0$ and $\hat{\boldsymbol{f}}$ denoting (layer-)normalized features.

For effective reasoning, each 3D Gaussian is assigned a semantic feature vector initialized from pre-trained 2D foundation models. We extract 2D features $F$ from these models and project them into the 3D Gaussian feature space using a learnable linear projection $W$:

$$V_{(l,t)}^{\mathcal{G}/\mathcal{F}} = W_V^{\mathcal{G}/\mathcal{F}} \cdot F_{(l,t)}^{\mathcal{G}/\mathcal{F}},$$

where $W \in \mathbb{R}^{d' \times d}$ is a projection matrix learned during training, $V \in \mathcal{V}$ is the graph node set. This strategy ensures that rich 2D semantic information can be smoothly embedded into each Gaussian representation. The overall construction procedure is summarized in Algorithm 1.

To embed these semantic features into the spatiotemporal graph, we build affinity matrices between node sets and apply Sinkhorn normalization to the full matrices (not element-wise). For two node sets $U$ and $V$ (e.g., within a frame or across two frames), let

$$\Phi_U = LN\big(W_e \, F_U\big), \quad \Phi_V = LN\big(W_e \, F_V\big), \quad S^{\mathcal{G}/\mathcal{F}} = \Phi_U \, \Phi_V^\top \in \mathbb{R}^{|U| \times |V|}. \tag{9}$$

We then obtain a (near) doubly-stochastic affinity by

$$A^{\mathcal{G}/\mathcal{F}} = \texttt{Sinkhorn}\big(S^{\mathcal{G}/\mathcal{F}}\big), \tag{10}$$

where $W_e$ is a learnable transformation, $LN(\cdot)$ is layer normalization, and $\texttt{Sinkhorn}(\cdot)$ denotes iterative row–column normalization over the entire matrix.

**Node-level Semantic Matching.** We align semantic features from the 3D Gaussian field ($\mathcal{G}$) and 2D foundation models ($\mathcal{F}$) by associating each node with a pair of projected feature vectors, $V_{(l,t)}^{\mathcal{G}}$ and $V_{(l,t)}^{\mathcal{F}}$, and define the node-level semantic matching loss accordingly.

$$\mathcal{L}_{node} = \sum_{V \in \mathcal{V}} \left\| V_{(l,t)}^{\mathcal{G}} - V_{(l,t)}^{\mathcal{F}} \right\|, \tag{11}$$

where $\mathcal{V}$ denotes the set of all graph nodes. The norm (commonly an $L_2$ norm) measures the discrepancy between the two representations at the same spatiotemporal location. This loss term compels the network to learn a 3D representation whose semantic characteristics are consistent with the high-level cues provided by the pre-trained 2D models.

**Struct-aware Semantic Matching.** For any pair of nodes at positions $(l, t)$ and $(l', t')$, we compute a normalized affinity that reflects their semantic similarity. Let $A_{(l \cdot t, l' \cdot t')}^{\mathcal{G}}$ and $A_{(l \cdot t, l' \cdot t')}^{\mathcal{F}}$.

denote the affinity values derived from the features of the Gaussian field and the 2D feature field, respectively. These affinities are computed after applying a learnable transformation (e.g., via a matrix $W_e$) and layer normalization ($LN(\cdot)$), and they are then normalized using the Sinkhorn algorithm to ensure proper distributional properties. The structure-aware matching loss is thus defined as

$$\mathcal{L}_{mat} = \sum_{A \in \mathcal{A}} \left\| A_{(l \cdot t, l' \cdot t')}^{\mathcal{G}} - A_{(l \cdot t, l' \cdot t')}^{\mathcal{F}} \right\|, \tag{12}$$

where $\mathcal{A}$ is the set of all affinity entries between node pairs. By minimizing $\mathcal{L}_{mat}$, we ensure that the pairwise similarity structure (i.e., which regions are similar or dissimilar) in the 3D Gaussian domain corresponds well with that in the 2D semantic domain. This structural consistency is essential for robust semantic propagation across both space and time, thereby benefiting downstream applications such as segmentation, tracking, and temporal alignment.

Table 1: Novel view rendering results on EndoNeRF Wang et al. (2022), Endovis17 Allan et al. (2019) and Endovis18 Allan et al. (2020) datasets.

| Methods | EndoNERF (*cutting*) | | | EndoNERF (*pulling*) | | | Endovis17 | | | Endovis18 | | |
|---|---|---|---|---|---|---|---|---|---|---|---|---|
| | PSNR ↑ | SSIM ↑ | LPIPS ↓ | PSNR ↑ | SSIM ↑ | LPIPS ↓ | PSNR↑ | SSIM ↑ | LPIPS ↓ | PSNR↑ | SSIM ↑ | LPIPS ↓ |
| NeRF Mildenhall et al. (2021) | 23.02 | 0.7930 | 0.3531 | 22.09 | 0.8106 | 0.3878 | 10.53 | 0.5448 | 0.4569 | 18.26 | 0.7716 | 0.3530 |
| EndoNeRF Wang et al. (2022) | 31.35 | 0.8913 | 0.1327 | 28.90 | 0.8573 | 0.1704 | 17.41 | 0.7472 | **0.3515** | 18.89 | 0.8028 | 0.3562 |
| NeRF-DFF Ye et al. (2023a) | 23.27 | 0.7117 | 0.4559 | 27.45 | 0.8326 | 0.2383 | 16.14 | 0.6464 | 0.3567 | 19.08 | 0.7857 | **0.3379** |
| 3DGS Kerbl et al. (2023a) | 22.79 | 0.7889 | 0.3857 | 21.03 | 0.8027 | 0.4313 | 18.74 | 0.7913 | 0.4125 | 18.93 | 0.7965 | 0.5231 |
| Deformable GS Lu et al. (2024) | 32.25 | 0.8756 | 0.1934 | 26.87 | 0.8532 | 0.2245 | 19.03 | 0.7935 | 0.4102 | 19.01 | 0.7978 | 0.5187 |
| EndoGaussian Liu et al. (2024b) | 34.10 | 0.9299 | 0.1209 | 28.94 | 0.8847 | 0.1433 | 19.12 | 0.7947 | 0.4028 | 19.06 | 0.8002 | 0.5173 |
| Feature DS Zhou et al. (2024) | 22.84 | 0.7909 | 0.3811 | 21.07 | 0.8027 | 0.4241 | 19.21 | 0.7959 | 0.4087 | 19.13 | 0.7981 | 0.5198 |
| 4DGS Wu et al. (2023) | 34.60 | 0.9360 | 0.1100 | 29.10 | 0.8850 | 0.1250 | 19.90 | 0.8150 | 0.3700 | 19.45 | 0.8080 | 0.5000 |
| GS-SLAM Yan et al. (2024) | 33.20 | 0.9240 | 0.1350 | 28.60 | 0.8800 | 0.1600 | 19.60 | 0.8100 | 0.3820 | 19.35 | 0.8050 | 0.5100 |
| EndoGS Zhu et al. (2024) | 33.60 | 0.9280 | 0.1280 | 28.70 | 0.8820 | 0.1550 | 19.30 | 0.8000 | 0.4050 | 19.20 | 0.8020 | 0.5180 |
| Free SurGS Guo et al. (2024) | 32.10 | 0.8720 | 0.2100 | 26.70 | 0.8500 | 0.2350 | 18.90 | 0.7920 | 0.4100 | 18.95 | 0.7980 | 0.5200 |
| LangSplat Liu et al. (2024a) | 31.80 | 0.8700 | 0.2200 | 26.50 | 0.8460 | 0.2400 | 18.80 | 0.7900 | 0.4150 | 18.90 | 0.7960 | 0.5250 |
| *SurstSplat* | **35.31** | **0.9424** | **0.0928** | **29.41** | **0.8887** | **0.0945** | **20.26** | **0.8205** | 0.3649 | **19.69** | **0.8120** | 0.4849 |

**Overall Optimization.** By combining these spatial interactions and temporal coherences graphs, our model effectively captures both spatial coherence within frames and temporal consistency across frames, leading to a more robust and holistic understanding of the surgical video. The photometric loss is defined as:

$$\mathcal{L}_{\text{rgb}} = (1 - \lambda)\mathcal{L}_1(I, \hat{I}) + \lambda\mathcal{L}_{\text{D-SSIM}}(I, \hat{I}), \tag{13}$$

combining $\mathcal{L}_1$ loss and structural similarity index (SSIM) to ensure both pixel-level accuracy and perceptual consistency.

Finally, our *SurstSplat* framework employs a multi-objective optimization strategy to build a robust 4D representation of surgical scenes. The overall loss function combines photometric and semantic consistency:

$$\mathcal{L}_{\text{total}} = \lambda_{node}\,\mathcal{L}_{node} + \lambda_{mat}\,\mathcal{L}_{mat} + \mathcal{L}_{\text{rgb}} \tag{14}$$

where $\lambda_{node}$ and $\lambda_{mat}$ are trade-off hyperparameters.

## 2.4 FEATURE-PROMPTED SCENE REPRESENTATION

We leverage foundation models as knowledge bases, distilling their 2D features into robust 3D scene representations. Using models like SAM and LLaVA-Med, we enable zero-shot or few-shot medical imaging tasks without task-specific training. Our teacher-student framework distills 2D capabilities—triggered by point, box, or text prompts—into the 3D domain, forming versatile feature fields. For a given target pixel, we assume an ordered set of $Q$ three-dimensional Gaussians,

$$\mathcal{Y} = \{\Theta_1, \Theta_2, \ldots, \Theta_Q\}, \tag{15}$$

where each $\Theta_i \in \mathcal{Y}$ corresponds to a 3D Gaussian kernel overlapping that pixel. For a prompt $\delta$, we compute an activation value on a Gaussian $\Theta$ using the cosine similarity between its feature embedding $h(\Theta)$ and a prompt query $q(\delta)$:

$$\varpi = \frac{h(\Theta) \cdot q(\delta)}{\|h(\Theta)\| \cdot \|q(\delta)\|}. \tag{16}$$

Let $\mathcal{S}$ denote a set of candidate labels (e.g., a text label set for semantic segmentation or a set of points for a point-based prompt). The probability that prompt $\delta$ corresponds to the Gaussian $\Theta$ is computed using a softmax function:

$$P(\delta \mid \Theta) = \frac{\exp(\varpi)}{\sum_{\varpi' \in \mathcal{S}} \exp(\varpi')}. \tag{17}$$

These probabilities are used to filter out Gaussians with low activation confidence. The selected Gaussians, after updating their corresponding color $c(\Theta)$ and opacity $\alpha(\Theta)$, participate in a point-based $\alpha$-blending process, enabling the generation of an edited radiance field from any new viewpoint. Consequently, the framework offers advanced functionalities, including immediate semantic segmentation without fine-tuning, identifying various tissues and structures in complex surgical settings. It also supports natural language commands for intuitive 3D scene editing and querying the 3D scene using natural language, enhancing surgical planning, guidance, and decision-making.

## 3 EXPERIMENTS

**Dataset and Evaluation.** We evaluate on four datasets: EndoNeRF, EndoVis18, EndoVis17, and EndoVis Conversations, covering tasks such as prostatectomy, instrument segmentation, depth estimation, tracking, and VQA. Each dataset uses its official train/val split; for EndoVis Conversations, we use 19,020 training and 2,151 test image-QA pairs, while the others follow an 80/20 split. Evaluation metrics include PSNR, SSIM, LPIPS for image quality, GPT-4 Score, accuracy, F-Score, mIoU for semantics, and training time, FPS, and GPU memory for efficiency; we also report feature PSNR (f-PSNR) based on mean squared error between predicted and teacher features. Concretely, f-PSNR is computed by treating the rendered and teacher feature maps as multi-channel images, computing the mean squared error over all channels and pixels, and then applying the standard PSNR formula; higher f-PSNR therefore indicates closer alignment of our 3D feature field with the 2D teacher features.

**Implementation Details.** We implement our method in PyTorch, using AdamW (lr=1e-4, cosine annealing, weight decay 1e-5), batch size 16, and 200 epochs. Data augmentation includes random cropping, horizontal flipping, and color jittering. Both $\lambda_{node}$ and $\lambda_{mat}$ are set to 0.1, and all experiments run on NVIDIA GPUs with monitored memory usage. We distill features from SAM (facebook/sam-vit-huge), CLIP (openai/clip-vit-large-patch14), and LLaVA-Med, projecting dense frame-wise features into the Gaussian space via a learnable linear layer. Camera intrinsics are assumed known; if poses are missing, we recover them with COLMAP (SfM) and optionally refine with GS-SLAM-style tracking Yan et al. (2024) for non-rigid sequences, following EndoN-eRF/3DGS conventions. To reduce teacher-model confounding, we use the same SAM-H and CLIP ViT-L/14 backbones for all feature-distillation baselines (NeRF-DFF, FeatureDS) wherever applicable. We further provide controlled ablations with different teacher capacities (SAM-B vs SAM-H) in the supplementary material, confirming that our relative improvements remain largely independent of teacher model strength. We will release the exact configuration files and checkpoints to facilitate fair comparison.

### 3.1 MAIN COMPARISON ON RENDERING QUALITY

Our method uses semantic features to generalize to unseen labels by mapping similar medical concepts to nearby regions in the embedding space. Distilling multimodal features for novel view semantic segmentation, we show significant improvements over standard Gaussian Splatting. Table 1 shows that *SurstSplat* outperforms baselines on all metrics across EndoNeRF_cutting, EndoNeRF_pulling, Endovis17, and Endovis18.

Table 2: Visual Question Answering (VQA) results by distilling Med-LLaVa. We report GPT-4 score for accuracy and FPS for efficiency.

| Metrics | Endovis17-C | | Endovis18-C | |
|---|---|---|---|---|
| | GPT-4 score ↑ | FPS↑ | GPT-4 score↑ | FPS ↑ |
| NeRF-DFF Ye et al. (2023a) | 76.89 | 42 | 74.57 | 69 |
| Feature DS Zhou et al. (2024) | 80.12 | 88 | 78.15 | 78 |
| *SurstSplat* | **82.97** | **123** | **81.25** | **114** |

Compared to NeRF-DFF, our model resolves the trade-off between semantic feature map quality and RGB fidelity, achieving higher accuracy across all metrics (PSNR, SSIM, LPIPS). For instance, on EndoNeRF_cutting, our method attains a PSNR of 35.31, SSIM of 0.9424, and LPIPS of 0.0928, outperforming all baselines. This results in superior visual quality and more precise segmentation masks

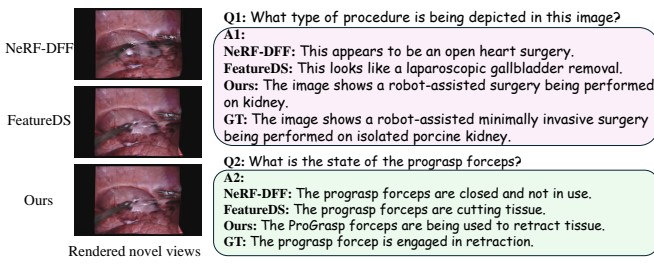

Figure 3: **Quantitative results of Medical Visual Question Answering (VQA):** Compared to NeRF-DFF and Feature DS, our approach yields precise and comprehensive responses.

Table 3: Temporal consistency on dynamic sequences. Higher t-SSIM and lower warped-LPIPS denote better temporal coherence.

| Methods | EndoNeRF (cutting) | | EndoNeRF (pulling) | |
|---|---|---|---|---|
| | t-SSIM ↑ | w-LPIPS ↓ | t-SSIM ↑ | w-LPIPS ↓ |
| 3DGS Kerbl et al. (2023a) | 0.880 | 0.180 | 0.865 | 0.200 |
| 4DGS Wu et al. (2023) | 0.930 | 0.110 | 0.915 | 0.125 |
| EndoGaussian Liu et al. (2024b) | 0.925 | 0.120 | 0.910 | 0.135 |
| EndoGS Zhu et al. (2024) | 0.920 | 0.130 | 0.905 | 0.145 |
| *SurstSplat* | **0.945** | **0.085** | **0.930** | **0.100** |

for both synthetic and real surgical scenes, enabling more accurate and detailed surgical scene reconstruction.

## 3.2 MEDICAL VISUAL QUESTION ANSWERING

We distill language-aligned 2D features (e.g., CLIP/LLaVA-Med) into the 3D field so that questions are embedded and matched to semantically consistent 3D regions across views and time. This improves entity localization and attribute retrieval in 3D, reducing ambiguity from single-view textures. Empirically, the graph-regularized features yield higher GPT-4 score and FPS than baselines that lack temporal semantic consistency. As shown in Fig. 3 and Table 2, our method achieves better VQA accuracy (GPT-4 score) and higher FPS across both Endovis17-C and Endovis18-C. Notably, our approach also demonstrates robust generalization to unseen question types and surgical scenarios, further highlighting its practical value in real-world medical applications. Following recent medical VQA works, we compute the GPT-4 score by prompting GPT-4 with the question, the ground-truth answer, and the model's answer and asking it to assign a discrete score in $\{0, 1, 2, 3, 4, 5\}$ according to a rubric (exact match, partially correct, incorrect); we report the average score over the test set, and provide the exact prompt and rubric in the supplementary material for reproducibility.

## 3.3 TEMPORAL CONSISTENCY EVALUATION

To assess temporal stability, we report two metrics in addition to frame-wise quality: (i) t-SSIM ↑, computed as the mean SSIM over pairs of adjacent rendered frames; and (ii) warped-LPIPS ↓, which measures LPIPS after warping the previous frame to the current one using forward optical flow (thus discounting explainable motion). Higher t-SSIM and lower warped-LPIPS indicate better temporal coherence without sacrificing detail. Results are summarized in Table 3. As shown in Fig 3, our method addresses the limited class diversity in medical datasets by leveraging advanced semantic features to map unseen labels to similar embedding regions, enhancing scalability and understanding of complex surgical scenes. Results in Tab. 2 show significant improvements in both segmentation accuracy and rendering speed compared to existing methods like NeRF-DFF and FeatureDS across multiple datasets. For example, on EndoNERF_cutting, our method achieves consistently higher IoU and Dice than NeRF-DFF, with reported values mutually consistent (Dice $= \frac{2\,\text{IoU}}{1+\text{IoU}}$).

## 3.4 NOVEL VIEW SEMANTIC SEGMENTATION

Our method addresses the limited class diversity in medical datasets through advanced semantic features. By mapping similar medical concepts to proximate embedding regions, the model effectively interprets unseen labels. This approach enhances scalability and comprehension of complex surgical scenes across various medical contexts. We distill multimodal features for novel view semantic segmentation in surgical environments. As shown in Table 4, our *SurstSplat* substantially outperforms NeRF-DFF and FeatureDS across all datasets, with higher IoU and Dice and faster rendering. These improvements, combined with strong generalization across surgical scenarios, demonstrate potential for enhancing computer-assisted surgery.

For scenes without official human annotations (e.g., EndoNeRF), we follow prior work and use SAM-generated masks as "Ground SAM" pseudo-labels to evaluate novel-view segmentation. This setup probes how well a 3D representation can propagate 2D segmentations to unseen viewpoints

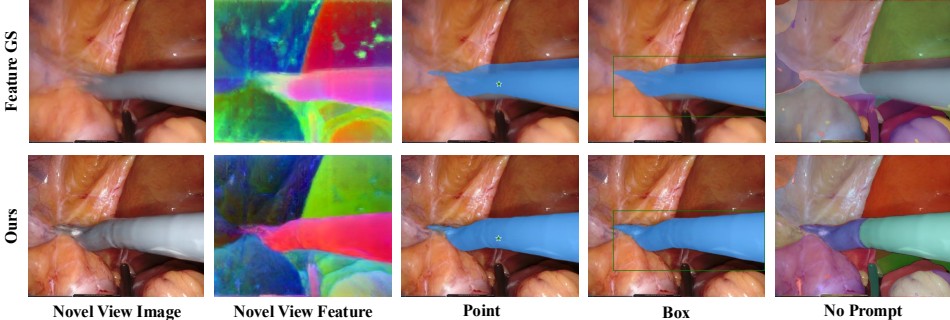

Figure 4: **Comparison of Novel View Segmentation results with Feature DS**: The Feature DS approach exhibits lower reconstruction quality and less precise segmentation masks. Our method achieves higher-quality masks, providing more detailed rendering.

Table 4: Segmentation results (Ground SAM) for rendered images from novel viewpoints.

| Metrics | EndoNeRF (cutting) | | | EndoNeRF (pulling) | | | Endovis17 | | |
|---|---|---|---|---|---|---|---|---|---|
| | IoU↑ | Dice ↑ | FPS↑ | IoU↑ | Dice ↑ | FPS ↑ | IoU ↑ | Dice ↑ | FPS↑ |
| NeRF-DFF Ye et al. (2023a) | 0.8258 | 0.0218 | 41 | 0.1543 | 0.0050 | 52 | 0.1961 | 0.1256 | 35 |
| Feature DS Zhou et al. (2024) | 0.9668 | 0.8993 | 98 | 0.9136 | 0.5422 | 85 | 0.5732 | 0.6644 | 110 |
| *SurstSplat* | **0.9945** | **0.9613** | **125** | **0.9449** | **0.9483** | **138** | **0.9963** | **0.7573** | **115** |

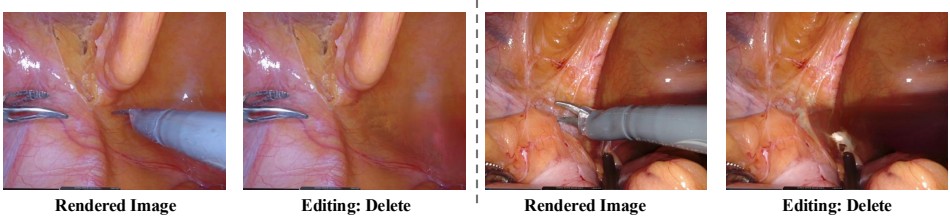

Figure 5: **Editing Results on the EndoNeRF dataset** : With the text of *"Delete tools"*, indicating our method's ability to accurately remove surgical instruments from rendered features.

rather than benchmarking absolute segmentation accuracy. On datasets with expert labels (EndoVis17/18), we additionally report metrics against the human-annotated masks in the supplementary material and observe consistent improvements, mitigating concerns about circular evaluation.

## 3.5 LANGUAGE-GUIDED EDITING

Figure 5 demonstrates our novel view editing capabilities, showcasing successful extraction and deletion of surgical instruments based on language inputs. Our approach achieves cleaner results with minimal artifacts compared to existing methods. The model's ability to selectively edit specific elements while preserving surrounding structures enhances surgical visualization and decision-making. This opens new possibilities for interactive surgical planning, intraoperative guidance, and advanced medical education tools that leverage comprehensive 3D surgical environment understanding from any viewpoint. Our method contributes to improving surgical precision, reducing procedural risks, and enhancing medical training, while establishing a foundation for intelligent surgical assistance systems that transform how surgeons interact with virtual representations of patient anatomy and surgical environment.

## 3.6 PROMPTABLE SEMANTIC SEGMENTATION

We present a promptable semantic segmentation approach based on the EndoNeRF framework Wang et al. (2022), enabling segmentation from novel viewpoints with flexible prompts such as points, boxes, or no prompts at all. As shown in Fig. 4, our method consistently outperforms the baseline

Feature DS, producing sharper and more detailed segmentation masks. The PCA-based feature visualization Pedregosa et al. (2011) reveals that our approach captures richer and more discriminative feature representations. For clarity, we report RGB rendering metrics (PSNR/SSIM/LPIPS) in Table 1 and semantic segmentation metrics (IoU/Dice) in Table 4, separating appearance fidelity from semantic accuracy.

### 3.7 ABLATION STUDIES

Table 5 presents an ablation study examining the impact of each model component. This analysis reveals the contribution of individual components to overall performance. The "No Temporal-aware" variant shows

Table 5: Ablation study on EndoNeRF dataset.

| Design Variants | Image | | | Feature |
|---|---|---|---|---|
| | PSNR↑ | SSIM↑ | LPIPS↓ | PSNR↑ |
| No Temporal-aware | 22.84 | 0.7909 | 0.3811 | 23.89 |
| No Semantics-embedded | 34.05 | 0.9269 | 0.1209 | - |
| No Spatiotemporal | 33.92 | 0.9301 | 0.1124 | 26.83 |
| No Graph Matching | 34.65 | 0.9336 | 0.1002 | 27.46 |
| Full Model | **35.31** | **0.9424** | **0.0928** | **28.65** |

a substantial decline in both image and feature quality, highlighting the importance of temporal awareness for dynamic surgical scene capture. For the "No Semantics-embedded" variant, while image quality remains reasonable, the model cannot generate effective feature representations, resulting in unmeasurable feature PSNR. Both the "No Spatiotemporal" variant (without spatiotemporal graph distillation) and the "No Graph Matching" variant exhibit decreased performance in image and feature quality. Our full model outperforms all variants across all metrics, confirming the effectiveness of each component and their collective contribution to system performance.

## 4 CONCLUSION

This work presents *SurstSplat*, a novel approach addressing dynamic 3D model reconstruction challenges in clinical settings. By combining multimodal feature-based Gaussian splatting with spatiotemporal graph distillation, *SurstSplat* effectively captures tissue deformations and enhances 3D visualization and instrument segmentation. Integrating knowledge from pre-trained foundation models improves real-time medical VQA performance. Our experiments confirm *SurstSplat*'s robustness and practicality in clinical environments, with significant potential for advancing robot-assisted surgery through its versatile adaptation to various tissue types and instruments. At the same time, *SurstSplat* has limitations: we do not explicitly model topological changes such as cutting or tissue removal, and we currently focus our evaluation on endoscopic datasets rather than generic dynamic scenes. We view extending our spatiotemporal semantic graph to handle explicit topology changes and validating the framework on additional surgical modalities and non-medical dynamic benchmarks as important directions for future work.

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

## A DECLARATION OF LLM USAGE

During the writing of the manuscript, we utilized a Large Language Model (ChatGPT) as a writing assistant. The scope of its usage was limited to **improving grammar, polishing sentences, and enhancing the clarity and fluency of this manuscript**. The method, claims, experimental results and conclusions are developed by the authors.

## B RELATED WORKS

**Traditional Methods for Surgical Scene Reconstruction.** Reconstructing 3D scenes from 2D images is crucial in surgical contexts. Conventional approaches such as Structure-from-Motion (SfM) in COLMAP Schönberger and Frahm (2016) and SLAM-based methods Zhou and Jagadeesan (2019); Song et al. (2017); Zhou and Jayender (2021) have been successfully applied to endoscopic reconstruction. Recent advances in Gaussian Splatting Kerbl et al. (2023b); Bao et al. (2024) represent scenes as optimizable Gaussian primitives initialized from SfM data, providing faster and more precise reconstructions. Enhanced frameworks like Gaussian-SLAM Yugay et al. (2023), GS-SLAM Yan et al. (2024), and SGS-SLAM Li et al. (2024b) further improve these methods, while specialized approaches such as EndoGSLAM Wang et al. (2024) tackle surgical-specific challenges including reflections and tissue deformations.

**Neural Rendering for Surgical Scene Reconstruction.** Dynamic surgical scene reconstruction is challenging due to tissue deformations from physiological processes and tool interactions. RoDyN-eRF Liu et al. (2023) addresses both static and dynamic radiance fields, while EndoNeRF Wang et al. (2022) models deformations with time-variant neural displacement fields. 3D Gaussian Splatting improves training efficiency and rendering quality, and methods like EndoGaussian Liu et al. (2024b) and DeformGS Duisterhof et al. (2024) further enhance flexibility and reduce computational costs. However, in-vivo reconstruction is still difficult due to limited tissue textures and variable lighting.

**Feature Field Integration from Foundation Models.** Feature field integration projects 2D visual features into 3D space to enhance reconstruction quality. Recent approaches like NeRF-DFF Kobayashi et al. (2022) transfer 2D feature similarities into 3D, while SA3D and GaussianGrouping Ye et al. (2023b) leverage foundation models such as CLIP Radford et al. (2021) and SAM Kirillov et al. (2023) to refine 3D masks. Our approach enables 3D Gaussian splatting with arbitrary-dimension semantic features through foundation model integration, improving rendering quality and enabling real-time applications. Recently, FeatureDS Zhou et al. (2024) integrated dense semantic features within the 3D Gaussian splatting framework. While this approach improved 3D reconstructions of static scenes, it struggles with dynamic surgical environments where tissue deformation and rapid movements create additional challenges.

## C METHOD

### C.1 SPATIOTEMPORAL GRAPH CONSTRUCTION

---

**Algorithm 1** Spatiotemporal Graph Construction

---

1: **Input:** frames $\{I_t\}$; Gaussians $\{g_{(l,t)}\}$; features $F$; hyperparameters $k$, temporal window $\Delta t$
2: **for** each frame $t$ **do**
3:     Build node set $U_t = \{v_{(l,t)}\}$; project features $V_{(l,t)}^{\mathcal{G}/\mathcal{F}} = W_V\, F_{(l,t)}^{\mathcal{G}/\mathcal{F}}$
4:     Connect $k$-NN spatial neighbors within $U_t$ to form $\mathcal{E}_{\text{space}}$ and compute weights $w_{uv}$
5: **end for**
6: **for** each frame $t$ **do**
7:     **for** each $t' \in [t - \Delta t,\, t + \Delta t]$ **do**
8:         Compute similarities $S_{U_t, U_{t'}}^{\mathcal{G}/\mathcal{F}} = \Phi_{U_t}\, \Phi_{U_{t'}}^{\top}$ and obtain affinities $A$ via Sinkhorn
9:         Connect top-$k$ pairs by $A$ to form temporal edges $\mathcal{E}_{\text{time}}$
10:     **end for**
11: **end for**
12: Compute $\mathcal{L}_{node}$ and $\mathcal{L}_{mat}$ using $V$ and $A$; update parameters by minimizing $\lambda_{node}\mathcal{L}_{node} + \lambda_{mat}\mathcal{L}_{mat} + \mathcal{L}_{rgb}$

---

