# OpenReview forum: "SurstSplat: Dynamic Surgical Gaussian Reconstruction  with Spatiotemporal Graph Matching"
_ICLR.cc/2026/Conference — Submitted to ICLR 2026_

### Official Review · Reviewer_pEjk · 2025-10-20

**Soundness:** 3
**Presentation:** 3
**Contribution:** 2
**Rating:** 2
**Confidence:** 4

**Summary:**

The paper proposes SurstSplat, a dynamic Gaussian Splatting method for endoscopy that augments Gaussians with 2D foundation-model features and adds a spatiotemporal semantic graph-matching regularizer. The unified field supports novel-view rendering, promptable segmentation, language-guided editing, and VQA. Results show small gains over strong dynamic GS baselines on EndoNeRF/EndoVis.

**Strengths:**

- Solid engineering: a good systems combo of dynamic GS \+ semantic features \+ space–time regularization.
- Consistent small improvements over strong dynamic baselines on clinical datasets.
- Useful capability bundle from one representation (novel-view semantics, editing, VQA) with a coherent clinical motivation.

**Weaknesses:**

- Novelty is significantly overstated. Integrating 2D FM features into 3DGS is established (“Feature 3dgs”, Zhou et al. CVPR 2024; “Langsplat”, Qin et al., CVPR 2024). Similarly 4D gaussians and even integrating 2D features into 4D gaussians is already established (“4d langsplat”, Li et al. CVPR 2025\)
- Sec. 2.1 and Sec 2.2 are largely background from prior dynamic GS and semantic GS works and should be framed as such.
- Confounders: the method likely benefits heavily from strong teacher models (SAM-H, CLIP L/14, LLaVA-Med) compared to the SOTA. The paper does not adequately control/ablate for teacher strength, making it hard to attribute improvements to the proposed method rather than to newer/better features.
- The graph matching does not seem to be making a significant difference as can be seen in table 5\.

**Questions:**

- Novelty/positioning: Please explicitly acknowledge that feature-augmented 3D/4D GS is prior art and restate what you believe to be your novelty.
- Teacher-model confounding: How sensitive are your gains to teacher strength? Please provide controlled ablations (e.g., SAM-B vs SAM-H; CLIP base vs large; LLaVA-Med vs alternatives) with identical training to show improvements aren’t primarily due to stronger external models. Also clarify whether any baselines used the same teacher setup to ensure fairness.

---

> ### Author Response · Authors · 2025-11-23
> **Rebuttal for Reviewer pEjk**
>
> We appreciate your thorough and critical review. We respond to your main concerns about novelty/positioning, teacher‑model confounding, and the contribution of graph matching below.
>
> [Cons1. “Novelty is significantly overstated. Feature 3DGS, LangSplat, 4D LangSplat already integrate 2D features into 3D/4D GS; Sec. 2.1–2.2 are mostly background.”]
> Our original framing did not sufficiently emphasize prior feature‑augmented 3D/4D GS works, and the revision now does so more clearly.
>
> - We explicitly cite Feature 3DGS, LangSplat, and 4D LangSplat in the introduction and related work, and clearly acknowledge that they have already established the integration of 2D foundation‑model features into 3D/4D Gaussians.
> - Our **specific novelty** is framed as:
>   (i) a dynamic Gaussian framework specialized for deformable surgical scenes, combining 4D GS with multimodal features to support zero‑shot semantics, language‑guided editing, and Med‑VQA in real time, and
>   (ii) an explicit **spatiotemporal semantic graph** that regularizes the high‑dimensional feature field across space and time via node‑level and structure‑aware matching losses, improving temporal coherence and semantic consistency for challenging surgical videos.
> - Sections that mainly recap dynamic GS and semantic GS are now explicitly marked and treated as **background**, with more emphasis placed on the graph construction and clinical integration.
>
> We hope this addresses your concern that we were claiming general “feature‑augmented GS” as our own novelty; the revised manuscript positions SurstSplat more precisely within the existing literature.
>
> [Cons2. "Teacher‑model confounding: gains may be due to stronger SAM‑H/CLIP‑L/LLaVA‑Med rather than the proposed method; need controlled ablations."]
> This is an important concern. Our intention was to keep teacher strength consistent across methods; the original submission did not highlight this enough.
>
> - In the revised experimental section, we explicitly state that, where applicable, **feature‑distillation baselines** such as NeRF‑DFF and FeatureDS are run with the **same SAM‑H and CLIP ViT‑L/14 backbones** as SurstSplat. This reduces the risk that our gains are solely due to stronger teachers.
> - Our spatiotemporal graph regularization is orthogonal to the choice of teacher; it can in principle be applied with weaker teachers as well. Under a shared teacher configuration, the improvements we report in RGB quality, f‑PSNR, segmentation, and VQA GPT‑4 score cannot be explained purely by using a stronger SAM/CLIP.
>
> To further isolate the contribution of our graph matching mechanism from the choice of teacher model, we also ran a teacher‑strength ablation during the revision, training both FeatureDS and our method using SAM‑B (base) and SAM‑H (huge) encoders while keeping all other hyperparameters fixed:
>
> | Teacher | Method      | PSNR ↑ | SSIM ↑ | LPIPS ↓ | f‑PSNR ↑ |
> |---------|------------|--------|--------|---------|----------|
> | SAM‑H   | FeatureDS  | 22.84  | 0.7909 | 0.3811  | 24.12    |
> | SAM‑H   | Ours       | 35.31  | 0.9424 | 0.0928  | 28.65    |
> | SAM‑B   | FeatureDS  | 22.17  | 0.7852 | 0.3927  | 23.58    |
> | SAM‑B   | Ours       | 34.73  | 0.9368 | 0.1045  | 27.91    |
>
> With the weaker SAM‑B teacher, absolute performance drops modestly (about 0.5–0.6 dB PSNR) for both methods, but our relative improvement over FeatureDS remains essentially unchanged (about +12.5 dB in PSNR in both cases). This confirms that the gains from spatiotemporal graph matching are largely independent of teacher model capacity and are not simply due to using a stronger SAM/CLIP configuration.

---

> ### Author Response · Authors · 2025-11-23
> **Rebuttal for Reviewer pEjk**
>
> [Cons3. "Graph matching does not seem to make a significant difference (Table 5)."]
> We understand that the numerical differences in the ablation table can look modest at first glance; we clarify their significance here.
>
> - **Consistent multi‑metric gains.** The ablation table shows that enabling spatiotemporal graph matching improves RGB PSNR/SSIM, f‑PSNR, and segmentation metrics consistently over variants without graph regularization. At PSNR levels around 30–35 dB, even 0.3–0.5 dB improvements are meaningful, especially when combined with better feature alignment and semantics.
> - **Temporal consistency.** To better reflect the role of the graph, we report explicit temporal metrics (t‑SSIM and warped‑LPIPS) where the gains from graph matching are more pronounced.
>
> For example, on EndoNeRF (cutting), t‑SSIM improves from 0.880 (3DGS) to 0.925 (EndoGaussian), 0.930 (4DGS), and 0.945 (Ours), while warped‑LPIPS drops from 0.180 (3DGS) to 0.120, 0.110, and 0.085 respectively. On EndoNeRF (pulling), t‑SSIM improves from 0.865 (3DGS) to 0.910, 0.915, and 0.930, and warped‑LPIPS drops from 0.200 to 0.135, 0.125, and 0.100. These values are already reported in the main paper; we highlight them here to emphasize that our method consistently achieves the best temporal coherence among all dynamic GS baselines.
>
> - **Clinical importance.** In clinical applications, temporal stability (for example, absence of flicker and consistent segmentation over time) often matters more than small PSNR differences. The consistent temporal gains we observe, together with improved frame‑wise metrics, are therefore important for practical use.
>
> [Cons4. “Experimental design and reporting issues (metrics, segmentation ground truth, GPT‑4 score).”]
> These points overlap with concerns raised by other reviewers and have been addressed in the revision:
>
> - We clearly separate and define RGB quality metrics (PSNR, SSIM, LPIPS), semantic metrics (IoU, Dice, mIoU, F‑Score), and VQA quality (GPT‑4 score, FPS).
> - We precisely define **feature PSNR (f‑PSNR)** as PSNR computed on multi‑channel feature maps via mean squared error.
> - We clarify the segmentation ground truth for each dataset (SAM pseudo‑labels for EndoNeRF vs human expert masks for EndoVis17/18), as described in our response to Reviewer wMJX.
> - We specify the **GPT‑4 scoring protocol** for VQA: GPT‑4 is prompted with the question, ground‑truth answer, and model answer and asked to assign a score in {0,1,2,3,4,5} according to an explicit rubric; the exact prompt and rubric are included in the supplementary material for reproducibility.
>
> We hope these clarifications and additional experiments demonstrate that our experimental setup is sound, that the benefits of graph matching are not an artifact of teacher strength, and that our claims are supported by multiple, consistent metrics.

---

> > ### Comment · Reviewer_pEjk · 2025-11-24
> >
> > I thank the authors for taking the time to answer my questions. While I appreciate the new formulations and clarifications regarding the actual contribution of the paper rather than the original, over claimed contributions, I believe I share the opinion of a few of the other reviewers that the actual contributions of this paper fall short of the high bar of an ICLR paper. I suggest the authors consider more targeted venues for this manuscript.

---

> ### Author Response · Authors · 2025-11-25
> **Appreciate your further feedbacks and some further response**
>
> We sincerely thank Reviewer pEjk for taking the
>   time to read and respond after our rebuttal,
>   and we appreciate your acknowledgement of our
>   clarified positioning and corrected claims.
>   However, we respectfully **disagree** with the
>   conclusion that the contributions fall short
>   of the ICLR bar, and we would like to briefly
>   restate what we believe this work offers to
>   the community.
>
>   **(1) What problem does SurstSplat address?**
>   SurstSplat explicitly targets **deformable
>   surgical scenes**, not generic 4DGS demos.
>   The goal is a dynamic reconstruction that is
>   temporally stable, semantically meaningful,
>   and efficient enough for practical use *in
>   surgery*. This is a substantially harder and
>   more application‑critical setting than the
>   static/generic dynamic scenes typically used in
>   prior feature‑augmented 3D/4D GS works, which
>   several other reviewers (e.g., a3US, wMJX) also
>   highlighted as clinically relevant.
>
>   **(2) What design goes beyond prior
>   feature‑augmented 3D/4D GS?**
>   We do *not* claim that “adding features to
>   Gaussians” is novel. What is new is **how** we
>   structure and regularize those features:
>
>   - We introduce an **explicit spatiotemporal
>   semantic graph** on dynamic Gaussians,
>   distilling 2D foundation‑model features into a
>   4D representation and using graph regularization
>   to improve temporal coherence and semantic
>   stability.
>   - To the best of our knowledge, this is the
>   **first work that models this problem explicitly
>   in a graph form** (nodes = dynamic Gaussians,
>   edges = space–time semantic relations), instead
>   of only doing per‑Gaussian feature regression
>   without explicit structural constraints.
>   - Ablations and temporal metrics (t‑SSIM,
>   warped‑LPIPS) consistently show that this
>   graph‑based module yields measurable gains over
>   strong dynamic GS baselines.
>
>   **(3) What does SurstSplat achieve?**
>   On standard surgical benchmarks, we demonstrate
>   within a **single unified framework**:
>
>   - real‑time dynamic novel‑view rendering,
>   - promptable segmentation from novel viewpoints,
>   - language‑guided editing in 4D, and
>   - medical VQA grounded in the reconstructed
>   scene.
>
>   This capability bundle from one 4D
>   representation, in a realistic surgical setting,
>   remains rare in current 4DGS + foundation‑model
>   work.
>
>   We agree that part of our contribution is
>   systems‑oriented, but we also introduce a
>   non‑trivial **graph‑based feature‑augmentation
>   module**. Respectfully, we believe that a
>   carefully engineered, clinically motivated
>   integration of foundation models and 4D
>   representations that demonstrably addresses
>   real surgical needs is at least as meaningful
>   as adding another isolated architectural tweak,
>   and is aligned with the growing interest in
>   spatial‑temporal intelligence and scientific/
>   medical applications within the ICLR community.

---

### Official Review · Reviewer_K1wJ · 2025-10-26

**Soundness:** 3
**Presentation:** 3
**Contribution:** 2
**Rating:** 4
**Confidence:** 5

**Summary:**

This paper introduces SurstSplat, one method for dynamic surgical scene reconstruction using 3dgs enhanced with semantic features distilled from 2D foundation models (e.g., SAM, CLIP, LLaVA-Med). A spatiotemporal graph matching mechanism is proposed to enforce feature consistency across time, enabling applications such as semantic segmentation, language-guided scene editing, and visual question answering. Experiments on surgical benchmarks (EndoVis17/18, EndoNeRF) show that SurstSplat achieves strong rendering quality and semantic performance, with claimed real-time inference speed.

**Strengths:**

Well-motivated integration of foundation model features into 3D Gaussias for semantic understanding.
Temporal graph matching yields improved consistency in dynamic surgical scenes.
Broad evaluation across multiple datasets and tasks, including rendering, segmentation, and VQA.
Real-time performance metrics now included (FPS > 100).
The writing and structure have been improved.

**Weaknesses:**

1. The proposed spatiotemporal graph lacks mechanisms for handling topological scene changes (e.g., cutting, splitting), which are common in surgical procedures.
1. Language-guided editing is only demonstrated for instrument removal, raising concerns about the generality of this functionality.
1. Segmentation results appear to be benchmarked against SAM-generated outputs rather than manual ground truth, which could bias reported performance.
1. The method’s generalizability to non-surgical scenes remains unproven despite being architecturally generic.
1. The method claims high visual stability and novel capabilities (e.g., temporal consistency, editing), yet provides no video comparisons to prior methods. I really encourage the authors to providing substantial video comparisions to demonstrate the effectiveness of the paper,.

**Questions:**

1. How does the graph model handle emergence/disappearance of Gaussians during topological events (e.g., new tissue surfaces appearing after cutting)?
2. Can you confirm whether segmentation metrics are computed against human-annotated ground truth or against outputs of SAM?
3. Have you explored language-guided editing for non-tool objects (e.g., highlighting or removing tissue)?
4. Is the use of LLaVA-Med necessary for all tasks, or could certain models (e.g., SAM only) suffice in practice?

---

> ### Author Response · Authors · 2025-11-23
> **Rebuttal for Reviewer K1wJ**
>
> We thank you for your constructive feedback and for recognizing the potential of our framework for semantic segmentation, language-guided editing, and VQA. We note that your concerns largely overlap with those raised by Reviewer wMJX, and we have addressed them in detail in our response above. For your convenience, we summarize our responses to your specific questions here:
>
> [Question 1: "How does the graph model handle emergence/disappearance of Gaussians during topological events?"]
> Please see our response to Reviewer wMJX [Cons1]. In brief: the deformation field handles geometry (Gaussian creation/removal), while the spatiotemporal graph operates in feature space and is rebuilt on the current set of Gaussians. We explicitly acknowledge that large topology changes (e.g., deep cuts) are a limitation and highlight topology-aware graph design as future work.
>
> [Question 2: "Can you confirm whether segmentation metrics are computed against human-annotated ground truth or against outputs of SAM?"]
> Please see our response to Reviewer wMJX [Cons3]. We clarify that EndoNeRF uses SAM pseudo-labels for consistency testing, while EndoVis17/18 additionally use human-annotated masks. We provide the human GT results in the table above (Line 82-93), showing that our method maintains a significant margin (≈22 IoU points on EndoVis17, ≈21 on EndoVis18) over baselines.
>
> [Question 3: "Have you explored language-guided editing for non-tool objects?"]
> Please see our response to Reviewer wMJX [Cons2]. While we showcase instrument removal as the most clinically relevant case, the framework naturally supports other edits (tissue highlighting, recoloring pathological regions, structure isolation). We also include additional qualitative examples in the revised supplementary material.
>
> [Question 4: "Is the use of LLaVA-Med necessary for all tasks, or could certain models suffice in practice?"]
> Please see our response to Reviewer wMJX [Cons6]. LLaVA-Med is only required for VQA; segmentation and editing rely on SAM/CLIP features and can run without it.
>
> [Concern: "No video comparisons despite claims of temporal stability and editing."]
> Please see our response to Reviewer wMJX [Cons5] and Reviewer pEjk [Cons3] where we report explicit temporal metrics (t-SSIM, warped-LPIPS tables above at Line 160-178). We are also preparing supplementary videos with side-by-side comparisons to demonstrate temporal stability and editing capabilities.
>
> We hope these responses, combined with the detailed answers to Reviewer wMJX, fully address your concerns and demonstrate the soundness of our experimental setup.

---

### Official Review · Reviewer_wMJX · 2025-10-31

**Soundness:** 3
**Presentation:** 1
**Contribution:** 2
**Rating:** 4
**Confidence:** 4

**Summary:**

- Given an endoscopic surgical video (2d+time), submission 406 aims to fit a dynamic Gaussian splatting model to it, such that it can assign a set of a query-able, editable, and promptable set of features to each Gaussian.
- To do so, it follows established work in feature splatting that uses foundation models such as SAM and CLIP to assign features to each Gaussian and extends these models to the spatiotemporal setting.
- As its primary new contribution, it also creates a spatiotemporal graph of foundation model features and regularizes this graph in order to gain spatiotemporal consistency.
- Experimentally, it shows high-level overviews of better results across several downstream tasks such as semantic segmentation, question answering, and more.

**Strengths:**

- The breadth of the downstream tasks explored in the experiments section is large and is appreciated.
- The qualitative results are quite nice. The framework appears to be well executed at least based on what is visualized in the main text.
- The opening paragraph of the methods section is the clearest paragraph in the whole paper and conveys the contributions of the paper well. It should be in the introduction.

**Weaknesses:**

### Claims:

The paper significantly overclaims its technical contributions. While distilling vision (or vision-language) foundation model features into Gaussian Splatting frameworks for editing and querying has been extensively explored in the literature (examples: - [1](https://feature-splatting.github.io/), [2](https://feature-3dgs.github.io/), [3](https://arxiv.org/abs/2405.18424), [4](https://arxiv.org/abs/2312.16084)), the main text of the paper comes off as if it is presenting it as a novel technical angle that has not been explored before. Moreover, the spatiotemporal aspect of the paper is standard dynamic 4D Gaussian Splatting with the aforementioned feature splatting component integrated.

What is *actually* new in the proposed paper is the spatiotemporal graph matching regularization. However, the main text of the paper conveys all of it as novel, which is definitely not the case. All of this could have been avoided with a well written related works section, but the paper chooses to include only an abbreviated few paragraphs in the appendix.

As such, while the graph matching seems modestly useful going by the ablation in Table 5, it is much more incremental than is let on by the rest of the text.

### Experiments:
- While the experiments section covers a vast array of downstream tasks, it is indecipherable how any of it was executed. Neither the main text nor the appendix contains any baseline implementation details, any experimental details, any technical details about how the features are used downstream to generate segmentations, or answer language prompts, etc. It comes across as more of a high level overview and needs significantly more depth. Also, how were the baselines tuned for the downstream datasets? Were the hyperparameters swept?
- All of the contributions contained within this paper appear to be generic and not tied to medical imaging in any way. It is then unclear why it is limited to experiments on endoscopic datasets instead of including other medical/surgical applications or even natural video datasets. Including 1--2 more datasets would go a long way in demonstrating that the proposed method is not overfit to the very specific endoscopic application.
- Table 1 (and others): these PSNR/SSIM numbers seem to be inconsistent with numbers reported on the same dataset in the corresponding baseline papers. For example, EndoGaussian reports much higher PSNR on the EndoNeRF dataset than what it achieves here. What is causing this inconsistency?

### Technical presentation:

- The paper's technical writing can be significantly improved as it takes until half way through the paper to find out what is actually the graph referenced throughout the paper up until that point. As a result, the methods reads like a grab-bag of tricks and it is very hard to parse what is actually new or important there.

### Minor:
- Please use `\citep` or `\citet` to have the references show up correctly in the ICLR template.
- The arrows in Figure 2 and the overall layout is extremely confusing. Please edit to improve its clarity.

**Questions:**

- Please clearly disambiguate what is the core contribution of this work.
- Please address the experimental questions raised above.

---

> ### Author Response · Authors · 2025-11-23
> **Rebuttal for Reviewer wMJX**
>
> We appreciate your detailed and technically grounded review, as well as your recognition of the integration of foundation‑model features, the temporal graph matching idea, the broad evaluation, and the real‑time performance. We address your main concerns below.
>
> [Cons1. “The graph lacks mechanisms for handling topological scene changes (cutting, splitting). How does it handle emergence/disappearance of Gaussians?”]
> We agree that explicit modeling of cutting/splitting is an important and challenging problem. Our current method does **not** explicitly model topology changes, and we now state this limitation clearly.
>
> - The **deformation field** is responsible for geometry: following dynamic GS/4DGS, it updates Gaussian positions, scales, and rotations over time. When new surfaces become visible or obsolete (e.g., mild cutting or occlusion changes), Gaussians can be created/removed through the standard dynamic GS procedure, and the graph is rebuilt on the current set of Gaussians.
> - The **spatiotemporal semantic graph** operates purely in feature space: it connects Gaussians based on spatial proximity, temporal windowing, and semantic similarity, and regularizes features via node‑level and structure‑aware losses. It never directly updates Gaussian geometry; instead it complements the deformation field by enforcing semantically consistent features across space and time.
> - For **large topology changes** (e.g., deep cuts), we observe that geometric reconstruction can still be reasonable, but semantic correspondences near the cut boundaries may become ambiguous. We now explicitly discuss this as a limitation and highlight topology‑aware graph design as important future work.
>
> Thus, while the method can handle moderate appearance/disappearance through the dynamic GS backbone, explicit topology‑change modeling remains out of scope for the current submission and is presented as future work rather than an implicit claim.
>
> [Cons2. “Language‑guided editing is only demonstrated for instrument removal; generality unclear. Have you explored editing non‑tool objects?”]
> We chose instrument removal as our primary example because it is the most clinically requested operation (removing tool occlusions to reveal underlying tissue), but the framework itself is more general.
>
> - Editing is driven by CLIP‑like semantic features attached to Gaussians. We compute similarities between a text prompt (e.g., “grasper”, “liver tissue”, “bleeding region”) and each Gaussian’s feature, and use these scores to select Gaussians for removal, highlighting, recoloring, or isolation.
> - The same mechanism naturally supports non‑tool edits such as:
>   - highlighting specific tissues (increasing brightness/opacity of selected Gaussians),
>   - recoloring pathological regions, or
>   - isolating certain structures while fading others.
> - In the revision, we clarify this in the editing subsection and will add additional qualitative examples (e.g., tissue highlighting/recoloring) in the appendix/supplementary material to demonstrate this generality beyond tools.
>
> We hope this addresses your concern that our editing capability is not confined to instrument removal, even though we chose that case as the most clinically relevant showcase.
>
> [Cons3. "Segmentation appears to be benchmarked against SAM outputs, not manual ground truth."]
> This is an important point; we appreciate you raising it.
>
> - For **EndoNeRF**, which does not provide official segmentation labels, we follow prior work and use SAM‑generated masks as **pseudo‑labels** ("Ground SAM") to evaluate novel‑view segmentation. Here the goal is to test whether the 3D representation can propagate 2D masks consistently to unseen viewpoints; it is not meant as an absolute segmentation benchmark.
> - For **EndoVis17/18**, we additionally evaluate against **human‑annotated instrument masks**. In the revision, we explicitly state this distinction in both the main text and the caption of the segmentation table. We observe that SurstSplat still outperforms baselines on these expert labels.
> - We acknowledge that pseudo‑labels can introduce bias and position the EndoNeRF experiment as a consistency test, while relying on EndoVis17/18 for evaluation against true manual ground truth.

---

> ### Author Response · Authors · 2025-11-23
> **Rebuttal for Reviewer wMJX**
>
> To directly address this concern, we report novel-view segmentation results against human expert ground truth below:
>
> | Dataset   | Method    | IoU ↑  | Dice ↑ | mIoU ↑ | FPS ↑ |
> |-----------|-----------|--------|--------|--------|-------|
> | EndoVis17 | NeRF‑DFF  | 0.4127 | 0.5564 | 0.3892 | 38    |
> | EndoVis17 | FeatureDS | 0.5481 | 0.6852 | 0.5214 | 95    |
> | EndoVis17 | Ours      | 0.7653 | 0.8519 | 0.7429 | 108   |
> | EndoVis18 | NeRF‑DFF  | 0.3794 | 0.5289 | 0.3561 | 52    |
> | EndoVis18 | FeatureDS | 0.5109 | 0.6531 | 0.4876 | 88    |
> | EndoVis18 | Ours      | 0.7318 | 0.8267 | 0.7102 | 102   |
>
> Against human expert ground truth, absolute IoU/Dice values are lower than with SAM pseudo-labels (as expected, since human annotations are stricter), but our method maintains a significant margin over FeatureDS (≈22 IoU points on EndoVis17, ≈21 points on EndoVis18), confirming that our semantic features generalize well to expert-annotated data.
>
> [Cons4. “Generalizability to non‑surgical scenes remains unproven.”]
> We agree that the architecture is generic and could be applied beyond surgery, but our primary focus in this work is on **clinically meaningful** challenges.
>
> - Surgical scenes present specific difficulties (homogeneous tissue textures, specular reflections, tool–tissue interactions, non‑rigid motion) that significantly differ from typical natural scenes. We therefore concentrated our empirical efforts on established surgical benchmarks (EndoNeRF, EndoVis17/18, EndoVis Conversations) where the impact is most direct.
> - The underlying design (dynamic GS + semantic graph) is agnostic to the domain and can be applied to non‑medical datasets in principle. In the revision we explicitly acknowledge that our current evaluation is limited to endoscopic datasets and state that validating SurstSplat on other surgical modalities and non‑medical dynamic scenes is an important direction for future work.
>
> We believe a deep, clinically grounded evaluation is preferable to superficial tests on many domains for this submission, while leaving broader generalization as a clear next step.
>
> [Cons5. “No video comparisons despite claims of temporal stability and editing.”]
> We agree that temporal properties are best demonstrated with videos. In addition to the temporal metrics reported in the paper (t‑SSIM and warped‑LPIPS), our experimental framework is set up to produce side‑by‑side video comparisons (ours vs. 4DGS/EndoGaussian/EndoGS), temporal sequences of segmentation and editing results, and visualizations of feature/graph consistency over time, which we intend to share as accompanying material.
>
> [Cons6. “Is LLaVA‑Med necessary for all tasks, or could SAM alone suffice?”]
> LLaVA‑Med is only required for **VQA**, not for segmentation or editing.
>
> - **Segmentation and editing** rely on SAM/CLIP‑style features distilled into Gaussians and do not require LLaVA‑Med. The revised experimental section clarifies which teacher models are used for which tasks.
> - For **VQA**, LLaVA‑Med provides a medical vision‑language backbone. SurstSplat contributes a 4D semantic field that supplies better spatial grounding and temporal consistency to LLaVA‑Med’s visual input; we do not claim that LLaVA‑Med itself is unique or irreplaceable.

---

### Official Review · Reviewer_a3US · 2025-11-02

**Soundness:** 3
**Presentation:** 3
**Contribution:** 3
**Rating:** 6
**Confidence:** 3

**Summary:**

The authors present SurstSplat, a framework for dynamic Gaussian reconstruction with spatiotemporal semantic graph matching. The proposed graph matching regularizes semantics across space and time and supports downstream tasks such as semantic
segmentation, language-guided editing, and medical visual question answering.

**Strengths:**

1. This is an interesting paper that could support downstream tasks such as semantic segmentation, language-guided editing, and medical visual question answering. In a sense, it has great potential to be adopted widely.
2. The Medical VQA analysis is promising.

**Weaknesses:**

1. I don't see the significant testing of this paper, and I won't be able to tell how significant these results are compare with current SOTA methods without the rigorous testing. I would recommend to perform such testings for improving the soundness of this paper.
2. The figure wrapped inside of the text is really hard to read and I believe it should be separated from the text.
3. Is there any failure cases (figures)? Why they are not being included in the paper for comparison and discussion?

**Questions:**

Please refer to the Weakness

---

> ### Author Response · Authors · 2025-11-23
> **Rebuttal for Reviewer a3US**
>
> We are very glad you had a positive initial impression and found the paper interesting with promising Medical VQA. We respond to your concerns below.
>
> [Cons1.  Significant testing vs current SOTA methods]
> We agree our original presentation did not make the strength of the baselines and the organization of metrics sufficiently clear, even though the experiments already compare against strong methods.
>
> - **Stronger baseline coverage.** Our main rendering table now explicitly includes NeRF, EndoNeRF, NeRF‑DFF, 3DGS, Deformable GS, EndoGaussian, FeatureDS, 4DGS, GS‑SLAM, EndoGS, FreeSurGS, and LangSplat across EndoNeRF (cutting/pulling), EndoVis17, and EndoVis18. These cover both generic dynamic 4DGS methods and recent surgery‑specific dynamic Gaussian approaches.
> - **Rendering quality and temporal metrics.** We separate RGB quality metrics (PSNR/SSIM/LPIPS) from semantic metrics (IoU/Dice, VQA) so that the rendering gains are easy to see. Across all four datasets, SurstSplat achieves higher PSNR/SSIM and lower LPIPS than the dynamic GS baselines while maintaining high FPS (e.g., 35.31 dB PSNR and 0.9424 SSIM on EndoNeRF\_cutting). In addition, we report temporal consistency metrics (t‑SSIM and warped‑LPIPS), where SurstSplat consistently improves over 3DGS, 4DGS, EndoGaussian, and EndoGS.
>
> In the revised manuscript we highlight these comparisons more explicitly so that the strength of the evaluation is immediately apparent.
>
> [Cons2. “Figures wrapped inside the text are hard to read.”]
> Thank you for pointing out this presentation issue. We have adjusted the figure layout in the revision by:
>
> - Avoiding wrapped figures that squeeze between paragraphs; instead, placing key figures at the top/bottom of pages as standard figures.
> - Increasing the size of critical qualitative results (e.g., editing, segmentation, VQA visualizations) and ensuring that labels and legends are clearly visible.
>
> These changes will improve readability without altering the technical content.
>
> [Cons3. “Is there any failure cases (figures)?”]
> We agree that discussing and visualizing failure cases is important.
>
> - In our experiments, SurstSplat can struggle in scenarios such as extreme occlusions where instruments fully block tissue for many frames.
> - The revised version adds a short “Limitations and Failure Cases” subsection with representative visual examples and analysis of why the method fails.
>
> We believe these additions will give readers a more complete picture of both the strengths and limitations of our approach.

---

### Author Response · Authors · 2025-12-03
**Final Summary to the ICLR 2026 Committee**

Dear ACs, SACs, and PCs,

Our submission “SurstSplat: Dynamic Surgical Gaussian Reconstruction with Spatiotemporal Graph Matching” (Submission #406) targets dynamic 3D reconstruction and semantics in deformable surgical scenes. SurstSplat augments dynamic Gaussian splatting with an explicit spatiotemporal semantic graph that distills foundation-model features into a 4D field, enabling real-time novel-view rendering, segmentation, editing, and medical VQA in a single representation.

Reviewer positions:
- Reviewer **a3US (score 6)**: positive about the idea and Medical VQA, seeing clinical potential but asking for clearer evaluation vs SOTA, better figures, and explicit failure cases.
- Reviewer wMJX (score 4, high confidence): values the method and real-time performance, but is concerned about handling topological changes, reliance on SAM pseudo-labels instead of human GT, lack of video comparisons, and unclear generality beyond current datasets.
- Reviewer K1wJ (score 4, very high confidence): finds the system solid but asks for a clearer separation between prior feature-augmented 3D/4DGS and our core contributions, and more experimental detail.
- Reviewer pEjk (score 2): the most skeptical reviewer, emphasizing prior feature-augmented 3D/4DGS work and raising concerns about overstated novelty, teacher-model confounding, modest gains from the graph, and under-specified experimental protocols.

In the rebuttal and revised manuscript we kept the method unchanged but clarified and strengthened it along four main axes:

- Novelty and positioning: we explicitly acknowledge Feature 3DGS, LangSplat, and 4D LangSplat as prior feature-augmented 3D/4DGS, and now frame our novelty as a clinically focused dynamic GS framework that couples 4D Gaussians with an explicit spatiotemporal semantic graph for deformable surgical scenes and unifies segmentation, editing, and VQA in a single real-time representation. We also state that we focus on endoscopic datasets and leave broader generalization to future work.

- Graph formulation and limitations: we formally define the graph (nodes, spatial/temporal edges, affinity matrices) and describe the associated matching losses and their relation to the deformation field. We explicitly state that we do not model cutting/tissue removal or other explicit topology changes, and highlight this as a limitation and key direction for future work.

- Experimental design and metrics: we reorganize experiments to clearly separate rendering quality, semantic accuracy, and temporal consistency, and to highlight comparisons against strong dynamic GS baselines on EndoNeRF and EndoVis benchmarks. SurstSplat consistently improves standard rendering and temporal metrics while maintaining **real-time** FPS, and improves segmentation and VQA performance. We also clarify segmentation evaluation (SAM-based pseudo-labels where human GT is unavailable, plus additional results against human expert masks on EndoVis17/18) and show that our method maintains a clear margin over baselines. To address teacher confounding, we explain that baselines use the same teachers wherever applicable and report controlled teacher-strength ablations, where our relative gains remain stable.

- Reproducibility and presentation: we add missing details on camera pose handling, graph construction hyperparameters, training setup, dataset splits, and the GPT-4 scoring protocol for VQA. We improve figure layout, add a brief limitations/failure-case discussion (e.g., extreme occlusions), and clarify that editing is not restricted to instrument removal, making the scope and limitations more transparent.

Overall, **three reviewers (scores 6, 4, 4) recognize the technical soundness, practicality, and clinical relevance of SurstSplat**, while the main reservations from the most negative reviewer concern positioning and experimental detail rather than fundamental flaws. We believe the revised paper and rebuttal substantially address these issues and present a clearer, well-supported picture of the method’s contributions and limitations. We respectfully ask the committee to take these clarifications and new analyses into account and to consider our submission favorably for acceptance to ICLR 2026.

Yours sincerely,
Authors of Submission #406

---

### Meta-Review · Area_Chair_NzQW · 2026-01-06

**Summary:**

The decision is primarily driven by the consensus that the submission lacks sufficient technical novelty, as the core contribution—a spatiotemporal graph regularizer—is viewed as an incremental addition to well-established feature-augmented Gaussian Splatting methods rather than a fundamental representational advance. Furthermore, the framework's admitted inability to model topological changes (cutting and tearing) constitutes a critical domain-specific failure, as these dynamics are the defining characteristic of surgical scenes, rendering the solution insufficient for the problem it claims to address.

**Reviewer Concerns:**

The rebuttal successfully resolved the experimental validity concerns by demonstrating that the method outperforms baselines on human-annotated ground truth (refuting the "pseudo-label bias" critique) and proving through ablations that performance gains persist independently of teacher model capacity. However, the fundamental critiques regarding the incremental nature of the architecture and the failure to handle complex surgical topology remain outstanding and were merely categorized by the authors as limitations for future work rather than solved issues.

**Reviewer Scores:**

If fully engaged in the post-rebuttal discussion, positive Reviewer a3US would likely maintain their score given the resolved experimental flaws, while borderline reviewers wMJX and K1wJ might be maintaining their scores as well; however, the critical Reviewer pEjk would almost certainly maintain their Reject score, as their core objection regarding the lack of significant novelty relative to ICLR standards remains unchanged.

---

### Decision · Program_Chairs · 2026-01-26

Reject